# AN ATTENTION FREE TRANSFORMER

## ABSTRACT

We introduce Attention Free Transformer (AFT), an efficient variant of Transformers (Vaswani et al., 2017) that eliminates the need for dot product attention. AFT offers great simplicity and efficiency compared with standard Transformers, where the multi-head attention operation is replaced with the composition of element-wise multiplications/divisions and global/local pooling. During training time, AFT has linear time and space complexity w.r.t. both the sequence length and feature dimension; in the autoregressive decoding mode, AFT has constant memory and time complexity per step. We show that, surprisingly, we are able to train AFT effectively on challenging benchmarks, and also to match or surpass the standard Transformer counterparts and other efficient variants. In particular, AFT achieves the state-of-the-art result on CIFAR10 autoregressive modeling with much reduced complexity, and also outperforms several efficient Transformer variants on Enwik8.

## 1 INTRODUCTION

Attention mechanisms, represented by Transformers (Vaswani et al., 2017), have driven the advancement of various machine learning problems, including language modeling (Devlin et al., 2018; Radford et al.), image modeling (Chen et al.), and set modeling (Lee et al., 2019). Different from other well known model architectures such as Convolutional Neural Nets (CNNs) or Recurrent Neural Nets (RNNs), Transformers enable direct interaction between every pair of elements within a sequence, which makes them especially powerful at capturing long term dependencies.

However, Transformers require high computational costs. The root cause of this challenge is the need to perform attention operations that have quadratic time and space complexity w.r.t the context size. This makes it especially difficult for Transformers to scale to inputs with large context sizes. A number of recent works have been dedicated to addressing the scalability issue of Transformers (Child et al., 2019; Kitaev et al., 2020; Rae et al., 2020; Wang et al., 2020b; Katharopoulos et al., 2020; Tay et al., 2020a; Choromanski et al., 2020). While the techniques adopted in the literature range from sparsity, locality sensitive hashing, low rank decomposition, kernel approximation and etc., most of them are trying to approximate the full attention operation.

In this paper, we take a bolder step towards the same goal, by proposing a computational module that does not use or approximate the standard dot product attention. We hence name our model the attention free transformer (AFT). Similar to dot product attention, AFT is composed of the interaction of three quantities, namely the query, key and value. What's different, however, is that AFT operates solely based on element-wise operations. To be more concrete, they key and value are first multiplied element-wise, the result of which is then pooled over the context dimension (in the causal model, this corresponds to a cumulative sum). The query is then multiplied with the reduced key-value representation element-wise to produce the final output. See Figure 1a for an illustration.

AFT maintains the full advantage of dot product attention, namely direct interaction between any two elements in a sequence (up to proper masking). However, the computational cost is drastically reduced to a $O(Td)$ complexity for time and space, where $T, d$ are the context length and feature dimension, respectively. In the autoregressive decoding mode, AFT also provides constant decoding time and space complexity per step, compared to $O(T)$ for standard transformers. To the best of our knowledge, AFT is the first model that achieves such efficiency in the context of Transformers. See Table 1 for the complexity analysis of AFT in comparison to other variants.

Table 1: Complexity comparison with different Transformers: Reformer (Kitaev et al., 2020), Synthesizer (Tay et al., 2020a), Linear Transformer (Katharopoulos et al., 2020) (only variants that support the causal mode are shown). Here $T, d$ denote the sequence length and feature dimension, respectively.

| Model | Time @ train | Space @ train | Time/step @ decode | Space/step @ decode |
|---|---|---|---|---|
| Full Attention | $O(T^2 d)$ | $O(T^2 + Td)$ | $O(Td)$ | $O(Td)$ |
| Reformer | $O(T \log T d)$ | $O(T \log T + Td)$ | $O(\log T + d)$ | $O(Td)$ |
| Synthesizer | $O(T^2 d)$ | $O(T^2 + Td)$ | $O(Td)$ | $O(Td)$ |
| Linear Transformer | $O(Td^2)$ | $O(Td + d^2)$ | $O(d^2)$ | $O(d^2)$ |
| AFT (ours) | $O(\mathbf{Td})$ | $O(\mathbf{Td})$ | $O(\mathbf{d})$ | $O(\mathbf{d})$ |

We show that we can interpret AFT as an extreme case of multi head dot product attention (MHA). In particular, we show that by 1) setting the number of heads equal to the feature dimension in MHA and 2) using $relu$ in place of $softmax$ as the non-linearity, MHA can be decomposed into the summation of two AFT modules (see Equation 6). However, this relationship is not true in a general sense, i.e., by varying the non-linearity injected after the query and key in AFT, we can obtain models that do not have a MHA counterpart. This realization allows us to freely explore the design choices (e.g., nonlinearity) of AFT to achieve the best performance. This philosophy is in direct contrast with previous and concurrent "linearized attention" works (Katharopoulos et al., 2020; Choromanski et al., 2020), which are constrained by the design space of MHA.

We perform experiments with AFT on several benchmarks, including unconditional image modeling, image super-resolution, language modeling, machine translation and point cloud generation. We show that AFT works very well as an alternative to the standard Transformer, providing competitive results as well as excellent efficiency.

To summarize, our contributions are as follows:

- We propose AFT, a new family of Transformer models that achieves $O(Td)$ time and space complexity in training, as well as $O(d)$ time and space complexity in autoregressive decoding.
- We show strong performance of AFT as a drop in replacement of MHA on various benchmarks, including setting the state-of-the-art result on CIFAR10 in the standard setting and outperforming other efficient Transformer variants.

## 2 MULTI-HEAD ATTENTION

At the core of Transformers is the Multi-Head Attention (MHA) operation. Given three sequences, namely the query $Q \in R^{T \times d}$, key $K \in R^{T \times d}$ and value $V \in R^{T \times d}$, and the number of heads $h$, MHA performs a scaled dot product attention for each head $i$, defined as:

$$f_i(Q, K, V) = \sigma(\frac{Q'_i (K'_i)^T}{\sqrt{d_k}}) V'_i, \text{ s.t. } Q'_i = QW_i^Q, K'_i = KW_i^K, V'_i = VW_i^V, \quad (1)$$

where $W_i^Q \in R^{d \times d_k}$, $W_i^K \in R^{d \times d_k}$, $W_i^V \in R^{d \times d_v}$ are linear transformations for head $i$, and $\sigma$ is the non-linearity by default set as the $softmax_r$ function (subscript $r$ indicates softmax is applied to each row of a matrix). $d_k, d_v$ are dimensions for key and value, respectively. MHA concatenates the output of $h$ attention heads along the channel dimension, resulting in feature dimension $hd_v$. Unless otherwise mentioned, we assume $d_k = d_v$ and $h = \frac{d}{d_k}$. This means the query, key and value are the same dimension within each head, and output dimension matches that of the input.

## 3 METHODOLOGY

### 3.1 ATTENTION FREE TRANSFORMER

We now define Attention free transformer (AFT), which provides an alternative to MHA. Given $Q, K, V$, AFT first linearly transforms them into $Q' = QW^Q$, $K' = KW^K$, $V' = VW^V$, then

performs following operation:

$$f(Q, K, V) = \sigma_q(Q') \odot \sum_{t=1}^{T} \big(\sigma_k(K') \odot V'\big)_t, \tag{2}$$

where $\odot$ is the element-wise product, with support for broadcasting when the operands' dimensions don't exactly match [1]; $\sigma_q, \sigma_k$ are nonlinearities applied to the query and key, respectively. Explained in words, the key and value are first combined with an element-wise multiplication, the result of which is then pooled over the context dimension, yielding a fixed length context vector $\in R^d$. This context vector is then multiplied with each row of the query, which forms the final output of an AFT layer.

One particularly useful variant of MHA is masked attention, oftentimes presented in the form of causal attention. Specifically, in auto-regressive models, queries are constrained to not be able to interact with keys and values beyond the curret position. In standard attention, this is usually implemented with an explicit binary masking matrix of shape $T \times T$, with non-causal entries masked as $0$. We show that it is also straightforward to extend AFT to the causal mode while maintaining its efficiency. We denote an AFT layer's output as $Y_t = f(Q_{\leq t}, K_{\leq t}, V_{\leq t}), \ t = 1, ..., T$ [2]. We formulate the casual AFT as:

$$Y_t = \sigma_q(Q'_t) \odot \sum_{t'=1}^{t} \big(\sigma_k(K'_{\leq t}) \odot V'_{\leq t}\big)_{t'}, \ t = 1, ..., T, \tag{3}$$

where the subscript $X_t$ indexes the $t$th row of matrix $X$.

**Discussions:** The design philosophy of AFT is to promote extreme efficiency, while keeping the benefit of standard Transformers. Concretely, AFT enables *direct* interaction of any two elements within the sequence, which is arguably the biggest advantage of Transformers over other types of models such as RNNs and ConvNets. However, AFT gets rid of the need of performing the costly spatial dot product attention, by computing a reduced value representation with the weights only depending on the keys. The resulting operation has an extremely efficiency of $O(Td)$ w.r.t. both time and space, which is the first model that achieves **linear complexity along both context and feature dimensions**. Moreover, the causal mode of AFT has an additional advantage of a constant decoding cost per step, similar to (Katharopoulos et al., 2020). To see this, from Equation 3, we have a simple recursion of $Y_t = \sigma_q(Q'_t) \odot (\sigma_k(K'_t) \odot V'_t + KV_{t-1})$ with $KV_t = \sum_{t'=1}^{t} \big(\sigma_k(K'_{t'}) \odot V'_{t'}\big)$, assuming $\sigma_q, \sigma_k$ are both element-wise functions. One thus only need to keep $KV_t$ in memory, and update it with constant cost per step.

**Selecting nonlinearies:** $\sigma_q, \sigma_k$ provide additional nonlinearity which helps to increase model's capacity. Empirically, we have found that one particularly strong setting is to let $\sigma_k = softmax$ which is normalized along the context dimension. This choice brings an interesting benefit especially in the causal mode, which we can explicitly write as:

$$Y_t = \sigma_q(Q'_t) \odot \Big(g_t(t) \odot V'_t + \sum_{t'=1}^{t-1} g_t(t') \odot V'_{t'}\Big), \ g_t(t') = \frac{\exp(K'_t)}{\sum_{t'=1}^{t} \exp(K'_{t'})}. \tag{4}$$

Here $g_t(t)$ acts as a role similar to that of an input gate in an LSTM, and $g_t(t')$ is operating like the forget gate, which depends on the input of time $t$, dynamically downweights the contribution of past time steps. When augmented with standard position embeddings as commonly used in Transformers, this allows the model to be able to learn the notion of recency at the same time of having access to the full context in the history. From this view, $\sigma_q$ can also be interpreted as the output gate, for which we found that both $sigmoid$ and $relu$ work well, with the former being slightly better. Also note that the same space and time complexity still holds for $\sigma_k = softmax$, both in training and decoding. In our experiments, unless otherwise mentioned, we use the $sigmoid + softmax$ setting for $\sigma_q$ and $\sigma_k$ by default.

**Relation to MHA:** Although AFT performs fundamentally different operations than standard attention, we show that the two family of models overlap in the extreme case. To see this, we explore

---

[1]We adopt Numpy styled broadcasting convention: https://numpy.org/doc/stable/user/theory.broadcasting.html

[2]We assume here that $Y_t$ includes input information at the current position $t$, the version where the current position is excluded can be obtained by shifting the outputs to the right.

the limit of number of heads in MHA, which amounts to letting $d_k = 1$ for each head. In this case, the dot product operation within each head reduces to a scalar product. Next, we set $\sigma$ to be $relu$ instead of $softmax$ in Equation 1. In this case, we have:

$$f_i(Q, K, V) = [Q'_i(K'_i)^T]_+ V'_i = ([Q'_i]_+[(K'_i)^T]_+ + [-Q'_i]_+[-(K'_i)^T]_+)V'_i$$
$$= [Q'_i]_+([K'^T_i]_+ V'_i) + [-Q'_i]_+([-K^T_i]_+ V'_i), \tag{5}$$

where $[\cdot]_+$ denotes the $relu$ operator, and $Q'_i, K'_i, V'_i \in R^{T \times 1}$ by definition. The concatenated output of the attention heads can then be concisely written as:

$$f(Q, K, V) = [Q']_+ \odot \sum_{t=1}^{T} ([K']_+ \odot V')_t + [-Q']_+ \odot \sum_{t=1}^{T} ([-K']_+ \odot V')_t, \tag{6}$$

which consists of two terms, each of which is an AFT operation, with $\sigma_q = \sigma_k = [\cdot]_+$ and $\sigma_q = \sigma_k = [-\cdot]_+$, respectively. However, note that this correspondence is not general, i.e., AFT does not need to approximate any MHA counterpart and can indeed have very different inductive biases than that of a standard Transformer.

**Relation to Linearized Attention:** There are a few recent works proposing to linearize the dot product attention (Linear Attention) from the view of kernel approximation, first proposed in Katharopoulos et al. (2020) and also in concurrent work (Choromanski et al., 2020). (Katharopoulos et al., 2020) proposes the linear attention operation in the form:

$$Y_t = \frac{\phi(Q'_t) \sum_{t'=1}^{T} (\phi(K'_{t'})^T V'_{t'})}{\phi(Q)'_t \sum_{t'-1}^{t} \phi(K'_t)^T}, \tag{7}$$

where $Q'_t, K'_t, V'_t$ are all row vectors of $R^d$. Equation 7 is similar to AFT, in the sense that the key and value are first combined and reduced in both cases. However AFT differs in two aspects: 1) the time complexity of Linear Attention is $O(Td^2)$, which is linear in the sequence length but has difficulty scaling to wide networks 2) Linear Attention is designed to approximate MHA, where the nonlinearity on query and key are shared. In AFT however, we show that it is beneficial to search for different nonlinearities for both the query and key.

### 3.2 LOCAL CAUSAL AFT

In autoregressive modeling, locality is a strong and effective inductive bias, as has been explored in Chen et al.; Child et al. (2019). We similarly propose an augmented version of causal AFT, where we rewrite Equation 3 as

$$Y_t = \sigma_q(Q'_t) \sum_{t'=1}^{t} w_{t,t'} (\sigma_k(K'_{\leq t}) \odot V'_{\leq t})_{t'}, \ t = 1, ..., T, \tag{8}$$

where $w_{t,t'} \in R$ is a locality masking scalar. We consider two strategies of constructing $w$, the first being the hard local mask where we have $w_{t,t'} = 1$ if $t - t's$ and 0 otherwise, with $s$ being the desired window size (for 2d inputs such as images, we can similarly construct 2d windows, see Appendix for details). The second one, which works better in practice, is to learn a position based local bias, while still assigning non-zero weights to out of window contexts. More concretely, we let $w_{t,t'} = \frac{\exp(I(t-t'<s)u_t^T v_{t'})}{\sum_{t'=1}^{t} \exp(I(t-t'<s)u_t^T v_{t'})}$, where $I(\cdot)$ is an indicator function, and $u, v \in R^{T \times d_u}$ are two sets of low dimensional learnable position embeddings, independently learned per layer. Note that in this case, we maintain dense connection between every $t, t'$ pair, but rather introduce learnable biases for more recent contexts. We typically set $d_u$ to be a small number (e.g., 64) which greatly reduces the amount of additional parameters compared to learning a full matrix. The learned version also adds very little overhead to both the time and space complexity, as $w$ is sparse and static, whose memory cost is marginalized out across batches during training. We denote the two versions as AFT-local-hard and AFT-local-learned, respectively.

## 4 RELATED WORK

Since the Transformer was introduced, there have been numerous attempts to address the major source of inefficiency in the architecture, the quadratic cost of the attention operation. Improving this

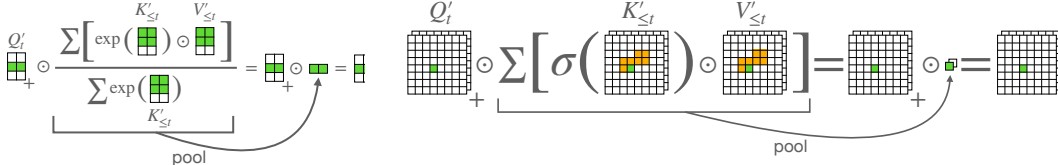

(a) Causal attention free operation.    (b) Local causal attention free operation.

Figure 1: AFT blocks require only element-wise and pooling operations.

operation can enable larger context sizes and more efficient implementations. For a comprehensive, recent survey of efficient transformers, see (Tay et al., 2020c).

**Approximating the dot product**. Katharopoulos et al. (2020); Choromanski et al. (2020) both propose to approximate the exponential kernel with inner product of projections, which leads to a linearized attention operation of complexity $O(Td^2)$. AFT is similar but offers greater efficiency of $O(Td)$ due to the exclusive use of element-wise operations, as well as more design flexibility. Reformers (Kitaev et al., 2020) apply LSH as an approximation to the dot product, whereas AFT completely gets rid the need of dot product.

**Sparse, local attention**. Sparse Transformers (Child et al., 2019) and Image Transformer (Parmar et al., 2018) proposes to use fixed sparse or local context patterns. Attention models in vision tasks (often combined with convolutions) use image structure to help handcraft relevant spatial patterns to attend (Wang et al., 2020a; Huang et al., 2019b; Zhu et al., 2019; Huang et al., 2019a; Ramachandran et al., 2019). AFT also borrows the locality idea, but we put it as a bias rather than hard constraint (see AFT-local-learned) . Also AFT is a standalone module, where it works as a plug in replacement of MHA in autoregressive tasks.

**Context compression**. Other approaches try to learn context patterns. Adaptive-Span Transformers (Sukhbaatar et al., 2019) learn a range for each attention head within which to attend. Routing transformers (Roy et al., 2020) use clustering to compute dot-product attention only over a subset of elements within the same cluster. The Linformer (Wang et al., 2020b) reduces the length of the context by compressing the keys and values with a linear layer. Compressive Transformers (Rae et al., 2020) compute and update reduced representations of the input that are far enough back in the input sequence, and attend to those compressed representations. AFT is largely complementary to these approaches, as our focus is to improve the complexity of any given sequence from the operation level.

**Eliminating dot product attention**. Instead of limiting the number of comparisons, other methods change the operation used to compute attention. The Synthesizer (Tay et al., 2020a) uses attention weights predicted from inputs, rather than derived from dot-product interactions. The LightConv module introduced in (Wu et al., 2019) proposes to replace the dot product self-attention with dynamic lightweight depthwise convolution, where the weights are normalized across temporal dimension. The Sinkhorn Transformer (Tay et al., 2020b) uses a differentiable sorting operation to identify relevant comparisons that may not be local in the original sequence order. AFT can be viewed as a more drastic version in this direction, where the we adopt a single global "attention mask" ($w$) of all ones (vanilla AFT) or with a few learnable entries (AFT-local-learned).

**Gated RNNs**. AFT is also related to the classic line of work on gated RNN variants, including LSTMs Hochreiter & Schmidhuber (1997), GRUs (Chung et al., 2014) and QuasiRNNs (Bradbury et al., 2016). AFT maintains the benefit of RNN models (linear complexity w.r.t. sequence length, constant decoding cost), but offers great parallelism and effectiveness, thanks to the use of a simple context reduction operation, which is amendable to a fully parallel implementation during training. We believe that AFT also offers new perspectives for rethinking the success and limitations of gated RNNs.

**Dynamic Convolution**. AFT is also related to dynamic convolution (Wu et al., 2019) when applied to auto-regressive tasks, where the reduced key-value representation can be interpreted as a per sequence convolutional kernel. However, AFT operates in an extreme case where the dimension of the kernel is 1 along both the feature and spatial dimensions, again presenting superior efficiency.

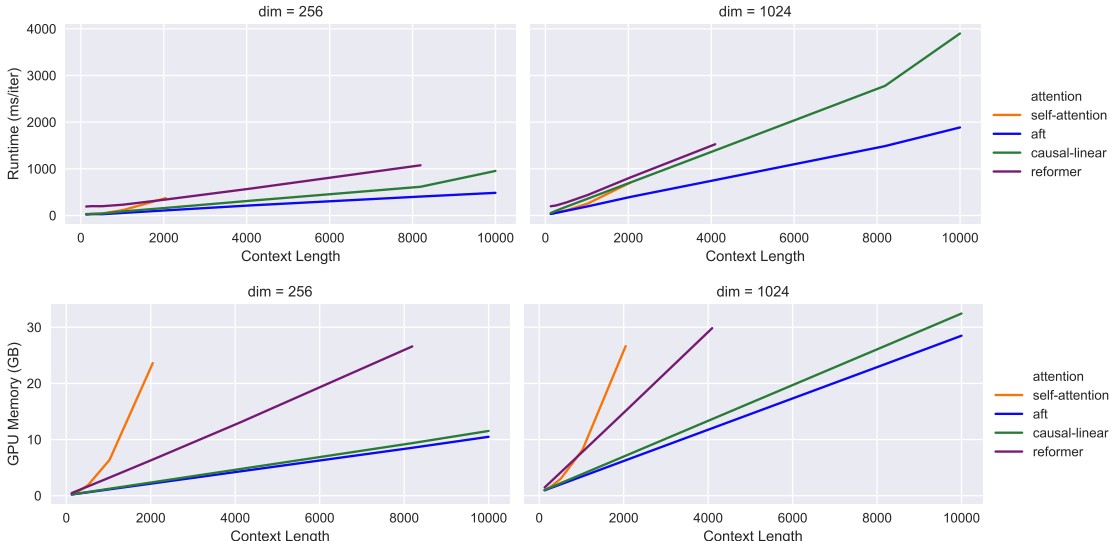

Figure 2: Comparisons of efficiency between models for a forward and backward pass with batchsize of 4 on a single GPU with 32 GB of RAM.

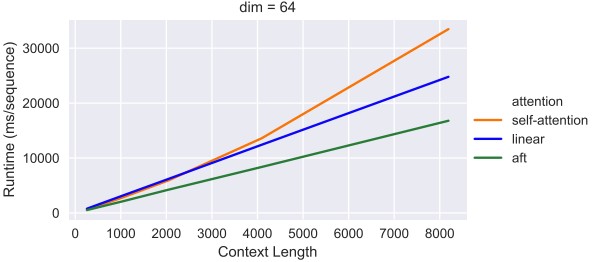

Figure 3: Comparisons of efficiency between models for decoding sequences.

## 5 EXPERIMENTS

We conduct experiments on five tasks: unconditional image modeling (Sec. 5.2), language modeling (Sec. 5.3), machine translation (Sec. 5.4), image super resolution (Sec. B.2) and point cloud generation (Sec. B.3). We focus on the causal mode of AFT, while leaving systematic evaluation of the non-causal version as future work. Unless otherwise mentioned, all experiments are conducted on $8\times$V100 GPU machines.

### 5.1 EFFICIENCY

To support our analysis in Table 1, we benchmarked an implementation of AFT on a single forward and backward pass on random data with a batch size of 4. We compared AFT with the self-attention from Transformers, a linear attention mechanism from (Katharopoulos et al., 2020), and a Reformer (Kitaev et al., 2020). For all of these, we used a base architecture with 12 layers, 8 heads (except for AFT, where there are no heads), feature dimension of either 256 or 1024, and context lengths up to 10,000. We used the code from the fast transformers library to perform the evaluations [3]. Results in terms of runtime and peak GPU usage are in Figure 2. Where data-points do not exist in the figure, the model exhausted GPU memory. We see from this that compared to Transformers and Reformers, AFT and linear attention require far fewer computational resources as context increases. In addition, we see that AFT is not as sensitive to the feature dimension as the linear attention, which expands the design space for feasible models. In all settings, AFT performs best. Additionally, we

---

[3]https://github.com/idiap/fast-transformers

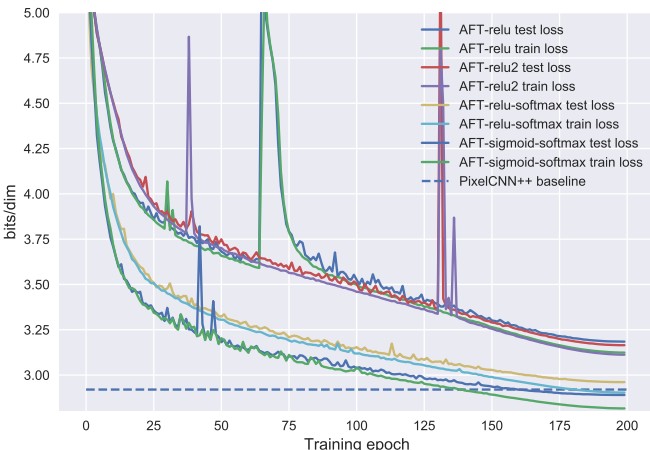

Figure 4: Proof of concept experiments for AFT-relu2, AFT-relu and AFT-softmax, tested on CI-FAR10. All three versions train well with standard optimization settings. AFT-relu2 and AFT-relu perform similarly, while AFT-relu-softmax and AFT-sigmoid-softmax are more stable and yields significantly better results.

examine decoding speed in the same benchmark. Figure 3, we see that the total runtime of AFT increases linearly with the context length, and is faster than linear attention and Transformers with self-attention.

## 5.2 UNCONDITIONAL IMAGE MODELING

In our first set of experiments, we consider the problem of image modeling by minimizing the negative log likelihood (NLL). Similar to Parmar et al. (2018), we represent an RGB image as a sequence of length $H \times W \times 3$, with $H, W$ being the height and width, respectively. Each sub-pixel is represented as a 256-way discrete variable. We use CIFAR10 for image density modeling.

**Feasibility study and choice of nonlinearities.** We first conduct experiments validating the legitimacy of AFT and its four nonlinearity variants. Our reference Transformer design largely follows that of Chen et al., where a transformer block consists of an attention layer (AFT layer in our cae) with residual connection and a 2 layer MLP with residual connections. Layer Normalization (LN) (Ba et al., 2016) is applied in a "pre-act" fashion. We adopt learned position embeddings, and use a set of shared token embeddings and prediction heads across RGB.

Our base architecture consists of 24 Transformer blocks, each with d=256 dimensional features. The hidden layer of the MLP per block has 4 dimensionality of its input. We use Adam, and follow a standard warmup learning rate schedule as in Vaswani et al. (2017). We use an initial learning rate of $3 \times 10^{-3}$ and a weight decay of 0.1 applied to all linear transformations weights, and a dropout of 0.1.

We adopt simple data augmentation. During training, we first randomly flip each image horizontally, then add or subtract a value in the range $[-10, 10]$ from all its subpixels, and clip resulting pixel values to $[0, 255]$. We use cross entropy loss, and a default batch size of 128 for 200 training epochs. We train three versions of AFT, namely AFT-relu2 (Equation 6), AFT-relu ($\sigma_q = \sigma_k = relu$), AFT-relu-softmax: ($\sigma_q = relu$, $\sigma_k = softmax$), AFT-sigmoid-softmax ($\sigma_q = sigmoid$, $\sigma_k = softmax$), all of which use full contexts. We show the training and test loss curves in Figure 4. All versions of AFT are trainable with standard optimization techniques for Transformers. In particular, AFT-relu2 performs slightly worse than AFT-relu, and both are significantly worse than AFT-relu-softmax and AFT-sigmoid-softmax. AFT-sigmoid-softmax also outperforms the strong PixelCNN++ baseline (Salimans et al., 2017). Based on this observation, we use AFT-sigmoid-softmax as the default setting for all remaining experiments, unless otherwise mentioned.

**Comparing with the state of the art.** CIFAR10 is a crowded benchmark for image autoregressive modeling, and we compare with a few competitive baselines, as shown in Table 2. Note that

Table 2: NLL results on CIFAR10, evaluated by bits/dim, the lower the better. Speed and memory are measured during training time, with a batch size of 32 across 8 V100 GPUs. AFT achieve the state-of-the-art result in this setting, with much fewer parameters, better speed and significantly less memory.

| Method | Train loss | Test loss | #params | Iters/Sec | GB/GPU |
|---|---|---|---|---|---|
| PixelCNN | 3.08 | 3.14 | | | |
| PixelCNN++ | - | 2.92 | | | |
| PixelSNAIL | - | 2.85 | | | |
| Sparse Transformer strided L=128, d=256 | - | 2.80 | 59M | | |
| Image Transformer local2d L=12, d=512 | - | 2.90 | 40M | 1.61 | 22.3 |
| AFT-full L=24, d=256 | 2.82 | 2.89 | 20M | **2.15** | **9.5** |
| AFT-local2d-hard L=24, d=256 | 2.81 | 2.87 | 20M | **2.15** | **9.5** |
| AFT-local-learned L=12, d=512 | 2.78 | 2.80 | 49M | 1.68 | 11.4 |
| AFT-local-learned L=24, d=256 | **2.75** | **2.74** | 29M | 1.67 | 12.8 |

CIFAR10 has an unrolled sequence length of 3072, which is already prohibiting to train a full Transformer with reasonable size. For example, for a standard 12 layer 512 dimension and 8 head configuration, the maximum batch size we can fit in our 8 V100 node is only 16, which makes it infeasible already. Our closest baseline Image Transformer (Parmar et al., 2018), which restricts attention to local2d windows of size of 256. We test our AFT-local-learned and aft-local2d-hard variants, with the same window size, and the same architecture as well as a deeper but narrower one (24 layer and 256 dimensions), which are still fair comparisons. We also compare to Sparse Transformers (Child et al., 2019), which restrains attention to sparse but global subset of context elements.

From Table2, we see that all AFT variants outperform the Image Transformer baseline. Both AFT local versions are better than the full counterpart, with AFT-local-learned being significantly stronger than others. We also observe that the deeper but narrower architecture is more effective than the shallow but wide baseline. Our best model also achieves the state-of-the-art result on CIFAR10 in this setting, outperforming a much larger Sparse Transformer model. Efficiency wise, we benchmarked Image Transformer against AFT variants on a 8 V100 GPU node [4]. All our variants are faster than Image Transformer, while consuming only half of the memory [5].

## 5.3 LANGUAGE MODELING

We apply AFT to character level language modeling on Enwik8 (Mahoney, 2011), which is another popular benchmark for auto-regressive modeling. We follow the standard preprocessing procedures and training/validation/test splits as in (Dai et al., 2019). Our base Transformer reference is a 12 layer 512 dimensional 8 head architecture with 2048 feed forward dimensions. For the first set of experiments, we use sequence length of 1024. Our training protocol as largely the same as before, except that we increase the weight decay to 0.5 and train for 100 epochs with batch size 128 in all experiments. We evaluate the AFT-local-learned variant with a window size of 32 and $d_u = 256$. We also compare to several efficient Transformer baselines, namely Reformer (Kitaev et al., 2020), Synthesizer (Tay et al., 2020a) and Linear Transformer (Katharopoulos et al., 2020). From Table 3, we see that with the base $L = 12, d = 512$ architecture, AFT achieves the lowest training bits per character (bpc), indicating its high capacity. Its test performance is slightly worse than that of the basic Transformer, but outperforms all other three variants. The deeper and narrower architecture of AFT strikes the best balance across parameter, speed, memory and performance. Its test bpc is only 0.02 away from the full Transformer's, while only consuming a third of the memory and provides a 44% speedup. In the end, we have also trained the same architecture with a sequence length of 2048, which results in an improved performance both on the training and test set. This suggests AFT's ability to effectively model long range dependencies.

---

[4] We use a batch size of 32 which is the largest batch size Image Transformer can fit

[5] Fair comparison against Sparse Transformer is infeasible, as it relies on a set of advanced implementation tricks such as mixed precision and gradient checkpointing, whereas AFT is implemented with standard Pytorch utilities ran in full precision.

Table 3: Enwik8 results, measured in bits per character (bpc), the lower the better. Baselines compared are Reformer (Kitaev et al., 2020), Synthesizer (Tay et al., 2020a) (it's best performing dense version) and Linear Transformer (Katharopoulos et al., 2020). Speed and memory are measured during training time, with a batch size of 128 on a 8 V100 GPU node.

| Method | T | Train bpc | Test bpc | #params | Iters/Sec | GB/GPU |
|---|---|---|---|---|---|---|
| Transformer L=12, d=512 | 1024 | 0.977 | 1.137 | 39M | 1.42 | 29.4 |
| Reformer L=12, d=512 | 1024 | 1.04 | 1.195 | 35M | 1.05 | 20.9 |
| Synthesizer L=12, d=512 | 1024 | 0.994 | 1.298 | 42M | 1.49 | 29.9 |
| Linear Transformer L=12, d=512 | 1024 | 0.981 | 1.207 | 39M | 1.46 | **10.6** |
| AFT-local-learned L=12, d=512 | 1024 | **0.854** | 1.18 | 45M | 1.85 | 11.3 |
| AFT-local-learned L=24, d=256 | 1024 | 0.973 | **1.157** | 32M | **2.04** | 11.2 |
| AFT-local-learned L=24, d=256 | 2048 | 0.942 | **1.139** | 45M | 0.97 | 21.9 |

**On the local window size.** In all our experiments, AFT-local-learned demonstrates superior performance compared to other variants. In order to validate its efficacy, we performed additional experiments wit the $L = 24, d = 256$ architecture, fixing everything but varying the local window size $s$. We show the results on 4, where we see that both the training and testing bpc forms a U-shape w.r.t. the window size, with 32 achieving the best performance.

Table 4: Training and testing bpc w.r.t. the local window size for AFT-local-learned.

| Window size | 0 | 1 | 2 | 4 | 8 | 32 | 64 | 128 | 256 | 512 | 1024 |
|---|---|---|---|---|---|---|---|---|---|---|---|
| Train bpc | 1.046 | 1.043 | 1.009 | 0.990 | 0.983 | **0.973** | 0.981 | 0.985 | 0.986 | 0.988 | 0.991 |
| Test bpc | 1.209 | 1.205 | 1.176 | 1.165 | 1.162 | **1.157** | 1.160 | 1.165 | 1.164 | 1.171 | 1.173 |

## 5.4 MACHINE TRANSLATION

As a machine translation benchmark, we show experiments with the WMT 2014 English to German translation task. The training set contains approximately 4.5 million sentence pairs. We compare against a Transformer architecture baseline using the OpenNMT implementation (Klein et al., 2017). For translation, the standard architecture is an encoder-decoder structure, where the encoder uses non-causal attention to encode the input sentence. The decoder uses two different types of attention. The first, self attention, sequentially attends to the output translation as it is being generated token by token. The second attends to the translation and the context from the encoder.

In our experiments, we replace the multi-headed decoder self-attention blocks. We compare perplexity (PPL), BLEU score, and efficiency between the Transformer base and AFT in Table 5. In this task, we see that AFT performs on par with the Transformer. As expected for the small context size, typically around 50 tokens, AFT does not show dramatic improvements in speed or memory.

Table 5: WMT 2014 English-to-German Translation.

| Method | Training PPL | Validation PPL | Test BLEU score | tokens/sec | GB/GPU |
|---|---|---|---|---|---|
| Transformer | 4.38 | 4.06 | 27.32 | 54.7K | 7.80 |
| AFT (Ours) | 4.34 | 4.10 | 27.70 | 54.4K | 7.54 |

## 6 CONCLUSIONS

We have introduced the Attention Free Transformer that replaces attention with an efficient, easy-to-implement new operation. We have demonstrated strong results on challenging benchmarks, despite of the simplicity of our design. We believe that our model opens a new design space for Transformer-like models, and will see impact in various areas where Transformers are applied.

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

## A    APPENDIX

## B    LOCAL AFT

---

**Algorithm 1:** Pseudo code of an efficient, in-place causal AFT-softmax/AFT-softmax-local1d.

---

**Input:** Query, key and value $Q', K', V' \in R^{T \times d}$; optionally context size $s \in \{2^n, n \in N\}$.
**Output:** Causal AFT output $Y \in R^{T \times d}$.

1  $KK = \exp(K')$ // new memory allocation
2  $KV = \exp(K') * V'$ // new memory allocation
    // if s is not provided default to $\lceil \log_2(T) \rceil$ iterations
3  **for** $j = 1, ..., \min(\lceil \log_2(T) \rceil, \log_2(s))$ **do**
4      stride $= 2^{j-1}$
5      KV[stride:, :] = KV[stride:, :] + KV[:T-stride, :] // in-place op
        // now $KV[i] = \sum_{k=max(0, i-2^j+1)}^{i} (\exp(K') * V')[k], \forall i$
6      KK[stride:, :] = KK[stride:, :] + KK[:T-stride, :] // in-place op
        // now $KK[i] = \sum_{k=max(0, i-2^j+1)}^{i} (\exp(K'))[k], \forall i$

    // normalize according to $softmax_c$ and multiply with query
7  $Y = relu(Q') * KV / KK$

---

**Algorithm 2:** Pseudo code of an efficient, in-place causal AFT-softmax-local2d.

---

**Input:** Query, key and value $Q', K', V' \in R^{H \times Wd}$, context sizes $s_h, s_w \in \{2^n, n \in N^+\}$
**Output:** Causal AFT output $Y \in R^{H \times W \times d}$.

1  $KK = \exp(K')$ // new memory allocation
2  $KV = \exp(K') * V'$ // new memory allocation
    // first aggregate locally across rows; pass if $s_w \leq 2$.
3  **for** $j = 1, ..., \log_2(s_w) - 1$ **do**
4      stride $= 2^{j-1}$
5      KV[:, stride:, :] = KV[:, stride:, :] + KV[:, :W-stride, :] // in-place op
        // now $KV[:, i] = \sum_{k=max(0, i-2^j+1)}^{i} (\exp(K') * V')[:, k], \forall i$
6      KK[:, stride:, :] = KK[:, stride:, :] + KK[:, :W-stride, :] // in-place op
        // now $KK[:, i] = \sum_{k=max(0, i-2^j+1)}^{i} (\exp(K'))[:, k], \forall i$

    // then aggregate locally across columns
7  **for** $j = 1, ..., \log_2(s_h)$ **do**
8      stride $= 2^{j-1}$
9      KV[stride:, :, :] = KV[stride:, :, :] + KV[:H-stride, :, :] // in-place op
        // now
            $KV[i_h, i_w] = \sum_{k_h=max(0, i_h-2^j+1)}^{i_h} \sum_{k_w=max(0, i_w-\frac{s_w}{2}+1)}^{i_w} (\exp(K') * V')[k_h, k_w], \forall i_h, i_w$
10     KK[stride:, :, :] = KK[stride:, :, :] + KK[:H-stride, :, :] // in-place op
        // now $KK[i_h, i_w] = \sum_{k_h=max(0, i_h-2^j+1)}^{i_h} \sum_{k_w=max(0, i_w-\frac{s_w}{2}+1)}^{i_w} (\exp(K'))[k_h, k_w], \forall i_h, i_w$

    // incorporate contexts to the right
11 idx = min(arange(W) + $\frac{s_w}{2}$, W-1) // arange(W) = [0, 1, ..., W-1]
12 KV[1:, :, :] = KV[1:, :, :] + KV[:H-1, idx, :] // in-place op
13 KK[1:, :, :] = KK[1:, :, :] + KK[:H-1, idx, :] // in-place op
    // normalize according to $softmax_c$ and multiply with query
14 $Y = relu(Q') * KV / KK$

---

### B.1    CIFAR10 VISUALIZATIONS

Here we show the visualizations of our best performing model trained on CIFAR10 (with test bits/dim 2.81). In Figure 5, we sample 32 test images and mask out the bottom half for each of them. We then use the model to sample the remaining pixels, one at a time. We see the model provides consistent completions for most cases.

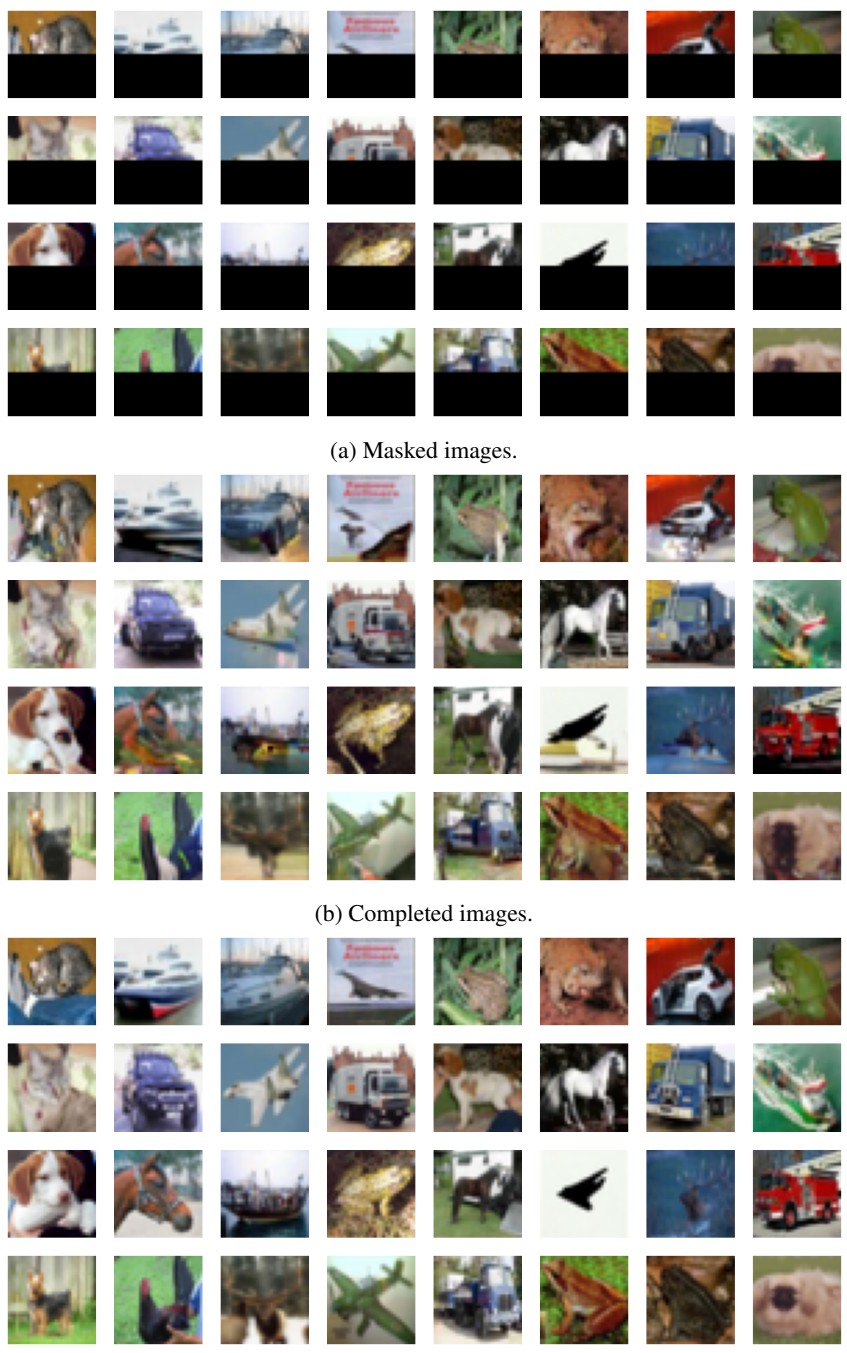

Figure 5: Image completion with test examples.

## B.2 IMAGE SUPER RESOLUTION

We also consider a super-resolution task based on pixel-wise image generation. Following (Dahl et al., 2017; Parmar et al., 2018), we enlarge an $8 \times 8$ sized image to $32 \times 32$.

We use CelebA dataset (Liu et al., 2015) as the benchmark. Our baseline model is the Image Transformer (Parmar et al., 2018) with its encoder and decoder connected through the attention mechanism. Both the 1D and 2D local Image Transformer models have $L = 12$ layers, $d = 512$ and

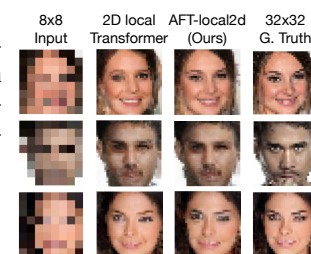

Table 6: Image super resolution results on CelebA. Our AFT models outperform the PixelRecursive baseline (Dahl et al., 2017) in bits/dim (the lower the better), and show clear advantages in parameter efficiency and memory saving over Image Transformers (Parmar et al., 2018), with comparable or even better performance.

| Method | L | d | Train bits/dim | Dev bits/dim | Params | Iters /sec | GB /GPU |
|---|---|---|---|---|---|---|---|
| PixelRecursive | - | - | - | 2.81 | 54M | - | - |
| 1D local Transformer | 12 | 512 | 2.54 | 2.68 | 39M | 2.98 | 9.5G |
| 2D local Transformer | 12 | 512 | 2.43 | 2.61 | 42M | 1.45 | 21.2G |
| AFT-local2d (Ours) | 32 | 256 | 2.39 | 2.59 | 25M | 1.43 | 10.4G |

Figure 6: Upscaled images from baseline and our 2D local transformers on CelebA.

attention heads=8, and are trained under the DMOL (discretized mixture of logistics) loss for 200 epochs. We experimented by replacing the standard attention blocks in decoder with our AFT-local1d or -local2d, keeping other modules the same. We follow similar training schemes, but with tuned dropout and learning rate. Evaluation is performed in terms of NLL measured in bits/dim, with the sampling temperature fixed at 0.8.

Table 6 shows results of our best AFT-local2d model (context size $16 \times 16$), in comparison to the PixelRecursive baseline and 1D/2D local Image Transformer models. Note the large consumption of model capacity and memory from 2D local Image Transformer, while our AFT-local2d shows clear advantages with no loss in model quality (see Figure 6 for visual comparison).

Figure 7 shows more samples from different models trained on CelebA face images.

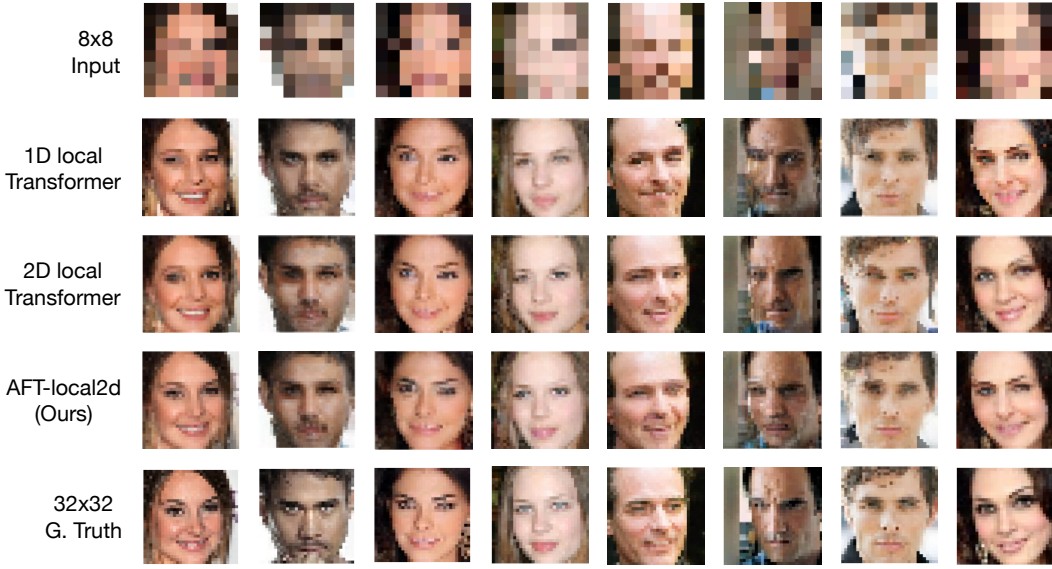

Figure 7: Upscaled images from baseline 1D/2D local Image Transformers (Parmar et al., 2018) and our AFT-local2D model trained on CelebA.

## B.3 POINT CLOUD GENERATION

In addition to images and text, we explore modeling point clouds randomly sampled from objects in the ShapeNetCore v2 dataset (Chang et al., 2015). Each point cloud consists of 2048 points. Following Nash et al. (2020), the points were sorted into a sequence in the order of z, y and x, then uniformly 8-bit quantized based on their positions. Our preliminary results of point cloud generation are shown in Table 7, and examples of generated point clouds are shown Figure 8. We see that our model is able to generate self consistent objects with fine details and great diversity.

Table 7: Results on ShapeNetCore v2, evaluated by bits/dim, the lower the better.

| Class | Train bits/dim | Test bits/dim |
|---|---|---|
| All classes | 3.67 | 3.87 |
| Airplane | 3.07 | 3.14 |
| Chair | 3.67 | 3.96 |
| Car | 3.64 | 3.99 |

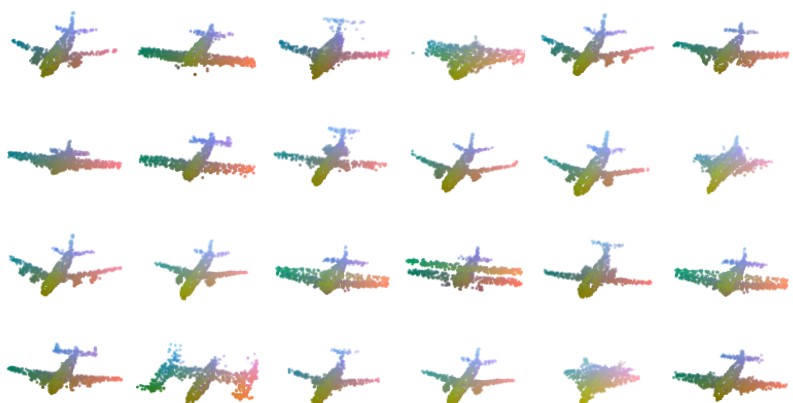

Figure 8: Point clouds generated by AFT trained on airplane point clouds.

