# OpenReview forum: "An Attention Free Transformer"
_ICLR.cc/2021/Conference — Reject_

### Official Review · AnonReviewer1 · 2020-10-27
**Hand-wavy connection to qkv-attention, missing baselines and no proper exploitation of increased efficiency.**

**Rating:** 4
**Confidence:** 4

**Review:**

The paper introduces a method to replace qkv-attention by a simpler, efficient building block. This is done by element-wise multiplication of a query representation with a compressed kv-memory. Per-channel attention pooling is used to compress the kv-memory. The model is derived from a softmax-free version of self-attention. The results show good performance on a couple of standard image and language modeling tasks while occasionally exhibiting favorable training speed. The results are largely on par with transformer baselines.

I have a couple of major concerns with this paper:

1) The derivation: The model is derived by using relus on the QK dot-product andthen simplified in Eq. 4, which basically makes the resulting model different from the starting point (Eq. 2-3). The final model is further changed by a applying a softmax over the keys, channel by channel (Eq. 5). So I am not sure how the resulting model actually still relates to the original formulation in Eq. 2. That leaves me with the impression that the derivation just exists to establish a connection with standard qkv-attention which is a bit hand-wavy. The results (Figure 2) even suggest that without a per-channel softmax over the input elements, the method doesn't work well. The arbitrariness of the derivation is exemplified further by the fact that one could have similarly started from using no non-linearity at all after the dot-product, which directly leads to Eq.4. The non-linearities on K and Q could be arbitrarily applied before the dot-product.

2) The efficiency comes at a cost of a strong memory bottleneck as we basically pool the entire memory into one fixed state on hidden size. That won't scale well to larger inputs. The conducted experiments are mostly on smaller scale settings (small images, standard text lengths) which reinforces my impression that this approach won't help solving the efficiency issues of self-attention. Using larger receptive fields (though costly) typically leads to better performance in standard self-attention. Here, however, due to the strong bottleneck results are getting worse after a point (Table 1). This makes also sense because the memory pooling imposes a very strong information bottleneck, that is, much of the memory has to be forgotten.

3) There should be at least some controlled baselines from related work that tries to eliminate the self-attention bottleneck with similar compression techniques, e.g. Linformer, Sinkhorn Transformer, Compressive transformer, Performers.

4) The potential efficiency gains are never put to practice, that is, the authors don't show any application of the model to very large input sequences.


Other comments and questions:

- The work reminds me of dynamic convolutions [1], which compute depth-wise convolution kernels dynamically based on the current context. Here we compute a dynamic depthwise 1x1 convolution on the Qs, based on the context around each Q. I think this connection might be more closely connected to the proposed model than attention.

- Why were the models not trained till convergence in on WikiText-103?


[1] https://arxiv.org/abs/1901.10430

---

> ### Author Response · Authors · 2020-11-25
> **Updated presentation and experiments**
>
> We thank the reviewer for the insightful comments.
>
> 1. Motivation and connection to MHA
> We first apologize for the confusion. Our intention is to propose AFT as a new family of model, not as an approximation to MHA. The "derivation" with relu and n_head=n_dim case is indeed trying to make an connection between AFT and MHA, showcasing the overlap of the two family of models. We have updated the draft, from the introduction thru methodology, to emphasize this view. As a result, the fact that the softmax version of AFT (which does not approximate any MHA counterpart) is an advantage, rather than a drawback of AFT. This property has also set AFT apart from other linearized attention works, such as Linear Attention [1] and Performers [2], which are constrained to design choices (e.g., non-linearities) that mimics a valid dot product attention. We also agree that one could also "derive" AFT from a linear attention (with dim=1). We choose to use the relu case as an example simply because it introduces two nonlinearities after Q and K, and it's also a reasonable baseline on it's own (it reaches the level of performance of a pixelcnn on cifar10).  Please refer to the updated draft for more clarification on this.
>
> 2. The pooling bottleneck.
> We agree that, intuitively the pooling operation may be a strong bottleneck for modeling long sequences. However, we do not agree with the claim that "this approach won't help solving the efficiency issues of self-attention". We respond to this from both the theoretical and empirical perspectives.
>
> Theoretically, it has been shown recently in [2] that the standard dot product attention can be approximated with Linearized attention methods with guarantees (see Theorom 4 in [2]), where the core of the linearized attention also corresponds to a pooling operation  along the spatial dimension. As a result, from the theoretical side, pooling is not necessarily a bottleneck compared to standard dot product attention.
>
> Empirically, we have demonstrated extremely promising results with the improved AFT (please see the updated experiments section). The notion of large context size is relative to many things. For example, the unrolled the sequence length of CIFAR10 is 3072 dimensional, which is a large input size for single machine training settings. To put this into context, on our 8xV100 GPU nodes, with a standard 12 layer 512 dim 8 head Transformer, we can only fit a maximum batch size of 16, which will take more than 4 days to train for 200 epochs with an optimized pytorch code. Also, one of the largest Transformer trained to date GPT3 [3] uses a context size of 2048, which is even smaller than that of CIFAR10. We thus believe that improved efficiency in our experimental settings have significant practical impact. Additionally, in our newly included benchmark Enwik8, we have shown that AFT is effectively able to gain from longer context sizes (last row of Table 3).
>
> 3. Baselines
> We have included comparisons with several baseline methods and outperformed them all. Please see Table 3 for more details.
>
> 4. Large sequence experiment
> Due to compute resource constraints, we do not evaluate super large scale sequence tasks. Our focus is instead on moderate (but practically challenging) sized task, and show improved efficiency and competitive performance. We do believe that our finds can be scaled to even larger scaled benchmarks, which we will leave as future work.
>
> 5. Relation to Dynamic Convolution
> AFT (as well as [1,2]) is indeed related to dynamic convolution, where the convolutional kernel size is 1x1. This extreme design offers AFT superior efficiency over dynamic convolution approaches. We have updated the related work section to include this reference.
>
> 6. Language model experiments
> We were not able to complete the language model experiments due to compute constraints. we have switched to Enwik8 instead and performed extensive experiments on it. Please check the updated draft for details.
>
>
> References:
> [1] Transformers are RNNs: Fast Autoregressive Transformers with Linear Attention, Katharopoulos et al.
> [2] Rethinking Attention with Performers, Choromanski et al.
> [3] Language Models are Few-Shot Learners.

---

### Official Review · AnonReviewer3 · 2020-10-28
**Interesting idea, motivation & evaluation require more effort**

**Rating:** 5
**Confidence:** 4

**Review:**

This paper proposes an efficient transformer variant by replacing softmax in the self-attention layer with a RELU activation and arranging the computation using element-wise products and global/local pooling. This reduces complexity to linear complexity in the non-autoregressive case and log-linear complexity in the autoregressive case. The evaluation shows that it can reach the performance of a vanilla transformer in most of the examined tasks while having fewer memory requirements in general.

**Strengths**:

The paper is reasonably well-written and clear for the most part. The problem of scaling transformers to longer sequences is an important one since transformers cannot deal otherwise with long sequences due to their quadratic complexity.

The proposed idea is interesting and reminiscent of recent methods that re-arrange self-attention computation using kernels albeit it differs in the way the computation is carried out. This one is simple computation-wise and does not aim to approximate the original computation in any way.

The evaluation performed on multiple tasks shows that the proposed approach can reach the quality of a vanilla transformer and be more memory-efficient.

**Weaknesses**:

(1) The motivation of AFT and the positioning with respect to prior work were somewhat weak. The introduction does not acknowledge recent efforts towards efficient transformers and what is the unique contributions of this work. What are the benefits of AFT compared to recent established efficient transformers such as Sparse Transformer (Child et al., 2019), Reformer (Kitaev et al., 2019), or Linear transformer (Katharopoulos et al., 2020)? It is unclear why one should prefer the proposed variants over existing ones both from theoretical and practical perspectives.  Related work states some previous efficient transformers without any individual discussion about their merits or limitations in comparison to AFT.

(2) One major limitation that stands out from the experiments, despite their size, is that there is no head-to-head or controlled comparison with a previously established efficient transformer such as the ones mentioned above. The results compared to Sparse Transformer given in Table 3 are not directly comparable since the model size and design are quite different.  In brief, it is not very clear what are the practical benefits compared to previous efficient alternatives.

(3) The memory benefits are not reflected or they are not as important when looking at the quality achieved in the tasks where a speed-quality trade-off was reported.  In language modeling,  AFT has higher perplexity (even when it uses a much larger number of parameters) which makes the memory benefits less interesting. In MT, AFT reaches the performance of the baseline but then the efficiency benefits are not present. So, I am curious is it the same in the two former tasks when comparing to the vanilla transformer? Under what circumstances we should expect AFT to reach vanilla transformer performance and still offer clear efficiency benefits when using the same setup?

(4) In terms of training speed, AFT is generally slower than the vanilla transformer when the form reaches the same quality as the latter. Also, it is especially slower when the depth is small in Table 2 (~30% with 12 layers). Could the authors elaborate a bit on why that happens?  Moreover, it would be useful to show in Table 2 what is the quality (NLL or bits/dim) achieved by each model because it's hard to tell how good the speed-quality tradeoff is.

(5) Recent studies have shown that it is possible to speed up inference time using efficient transformers (see above). What is the benefit of AFT during inference time?

---

> ### Author Response · Authors · 2020-11-25
> **Thanks for the insights and please see the updated paper**
>
> We thank the reviewer for the insightful comments and suggestions, which have been absorbed into the updated draft.
>
> We agree with every point of your review. AFT is indeed related to linearized attention but it's more simple and efficient and is also not trying to approximate the standard MHA. We have made this point more explicit in the updated draft. On the specifics:
>
> 1. Positioning and comparison to related work
> We agree that the original version did not this clear, which might lead to confusion and misunderstanding. We have made substantial updates to the introduction and presentation of AFT, including addressing existing works in the Introduction section, adding Table 1 to illustrate the benefits of AFT, a reorganized related work section with direct comparisons with groups of related  works, as well a new methodology section highlighting the uniqueness of AFT and its relationship to MHA, linear attention and gated RNNs. We have also included Figure 2&3 as empirical counterparts to Table 1. Please check the paper for details.
>
> 2. Control experiments with other variants
> We have also included many new experimental results to address this concern.
> ---On CIFAR10, we have benchmarked the local2d version of Image Transformer wrt speed and memory usage. We show that all AFT variants outperform Image Transformer, oftentimes with large margins and being much more efficient.  We have also included more discussions regarding this.
> ---We have introduced a new benchmark Enwik8, with which we evaluated AFT against Linear transformer, Reformer and Sythesizer. We show that AFT outperforms all these variants w.r.t. both performance and efficiency.
>
> 3. Memory efficiency.
> We have addressed this concern with updated, stronger results, and our optimized implementation which results in greater efficiency in practice. To summarize, on both CIFAR10 and Enwik8, we have demonstrated superior memory and speed efficiency while matching or outperforming the standard Transformer baseline. On MT, our optimized code yields similar speed efficiency compared to standard transformer but slightly better memory footprint. Figure 2&3 can also help to address this question.
>
> 4. Training speed.
> Our previous code is heavily under optimized, and we now have a much improved speed, faster in most cases than standard and other efficient Transformer variants. The paper has been updated with these new results. We have also merged the original Table 2 into the new Table 2, where we show speed and memory of AFT variants. For the full Transformer variants, we unfortunately could not afford to train them in a reasonable amount of time, so we instead benchmarked the local2d version of Image Transformer.
>
> 5. Decoding efficiency.
> AFT has constant decode cost per step, which is advantageous over standard and many other Transformer variants. See Figure 3 in the updated draft.

---

### Official Review · AnonReviewer4 · 2020-10-28
**Replacing the soft-max in the Mult-Head Attention operation with "relu" for Transformers (O(T^2) -> O(T))**

**Rating:** 6
**Confidence:** 3

**Review:**

The paper suggest an alternative to the Multi-Head Attention (MHA)  operation, which is one of the core elements in Transformers models. The proposed alternative is targeting the non-linear soft-max operator (in the MHA) and suggest to replace it with the "relu" operator. After doing so they could reformulate the new attention mechanism as a O(T) operator instead of the original O(T^2) operator (where T is the context size).

Arguably, the MHA is one of the important components of the transformers architecture and reducing its memory and time complexity is crucial to increasing the training batch-sizes and the usage of more context.

The paper is nicely written and presents a comprehensive experimentation section, ranging over several machine learning benchmarks in computer vision and NLP.

Strong points:
- simple solution
- comparable results with reduced memory and latency
- the paper is clear and nicely written
- comprehensive experimentation section

Weak points:
- moving from AFT-relu to AFT-softmax is is not sufficiently motivated (only empirically) i would expect more experimentation to clear this point
- not all experiments show improved or comparable results (for example table 5)

---

> ### Author Response · Authors · 2020-11-25
> **Updated presentation and results**
>
> We thank the reviewer for the positive feedback as well as insightful suggestions.
>
> 1. Motivation of AFT.
> Thanks for raising this concern, and we have provided a significantly updated presentation of AFT. In our new draft, we have emphasized that we present AFT as a new family of model, which drawing connections to MHA, Linearized Attention, as well gated RNNs. Please see the updated introduction, methodology as well as the related work sections for more clarification on this.
>
> 2. Experiments
> We have included substantially improved experimental results, including stronger and state-of-the-art results on CIFAR10, a new inclusion of Enwik8 and extensive experiments, and also improved benchmarking results on machine translation due to our optimized implementation. Please refer to the experiments section for more details.

---

### Official Review · AnonReviewer2 · 2020-10-30
**A replacement to multi-head self-attention operation with vague usefulness**

**Rating:** 4
**Confidence:** 5

**Review:**

This paper introduces the Attention Free Transformer (AFT), an alternative to multi-head attention (MHA) operation in Transformer. While the motivation of the authors is to replace MHA with more cost-efficient operation, it is not clear whether the proposed method is the better alternative.

Pros:
1. AFT shows better asymptotic space and time complexities than MHA.
2. The implementation of AFT allows for faster training with larger batches.

Cons:
1. Theoretical analysis is conducted for the extreme case of num_heads=hidden_dim and ReLu non-linearity. It is not clear how to generalize them to more practical cases with num_heads<hidden_dim and SoftMax non-linearity. There is a missing link between the theory (and motivation arising from it) and the best-performing implementation (AFT-softmax).
2. AFT-softmax does not fully complies with the title of the paper as the proposed operation contains aggregation via softmax. Also, despite the claim that AFT can "be readily adopted as a plug in alternative to Transformers", the architectures from the experimental section also use vanilla MHA blocks in addition to AFT. Thus, it is an exaggeration to say that the Transformers evaluated in the paper are attention-free.
3. Language modeling experiments on WikiText-103 draw an ambiguous picture. Baseline Transformer implementation has large positive difference between train and val/test PPL at 70k iterations which decreases as the training progresses. For AFT models, on the other hand, this difference is negative which might suggest that they have already overfit at 70k iterations and they will never reach the resulting performance achievable by the baseline. The plot with train&val PPL / number of iterations for those experiments would be more informative than the table.

---

> ### Author Response · Authors · 2020-11-25
> **Presentation updated, more experiments added**
>
> Thank you for pointing out the concerns and insightful comments.
>
> 1. relation with MHA
> We propose AFT as a new family of model, rather than merely an approximation to MHA. The core operation of AFT is very different from a standard MHA in most cases, but there is also a connection in the case of ReLU nonlinearity and n_heads=n_dim. As a result, AFT has its own design space, and does not have or need to have the notion of multi heads anymore.  We apologize for the confusion, and have made substantial changes to the  presentation. Please check the Methodology section of the updated drafts for more details.
>
> 2.1 attention free is exaggeration
> We again apologize for the confusion. We refer to attention in our paper as the standard "dot product attention", which is implied but not explicitly stated. The softmax (denoted as \sigma_k in the updated draft) in AFT can indeed be viewed as some form of "attention", but it's implication is drastically different as it does not involve a dot product and it has O(Td) complexity rather than O(T^2d). We have updated the draft bu making the reference to dot product attention more explicit.
>
> 2.2 AFT as drop in replacement to MHA
> Our experiments have been dedicated to evaluating the causal mode of AFT, similar to Linear Attention [1], while we leave the evaluation of the non-causal mode as future work. That being said, in both the image autoregressive modeling and language modeling experiments, AFT is the only building block used. In our updated draft, we have shown strong performance on both two benchmarks, including achieving the state-of-the-art result on CIFAR10, as well as outperforming several other efficient Transformer variants.
>
> 3. The initial results on wikitext103 were not complete in time and prematurely included. We were not able to perform extensive experiments on this dataset due to compute recourse constraints. We have thus decided to switch to the Enwik8, a popular character level language model benchmark. We show that AFT yields competitive performance and excellent efficiency. Please check the updated draft for details.
>
> References: [1] Transformers are RNNs: Fast Autoregressive Transformers with Linear Attention, Katharopoulos et al.

---

### Author Response · Authors · 2020-11-25
**Draft Updated**

We thank all the reviewers for the insightful comments and suggestions. We have provided significant updates to the model, presentation and experiments, by incorporating the feedbacks from the reviewers. We summarize the major updates here and encourage all reviewers to check the paper for details.

*****Improved model design*****
We have made small changes to our model design which leads to consistent improvements of our model's quality. Two important changes are:
1. We have found that using sigmoid instead of rely for Q leads to constant gains (\sigma_q in Equation 3).
2. We have introduced a new variant of local causal AFT which becomes our best performing variant, see Section 3.2 for details.

*****Improved Implementation*****
We have also made significant progress optimizing our training code, resulting in much improved efficiency, especially wrt to training time. All results are updated to reflect this change, and we recommend the reviewers to check the paper for details.

*****Improved presentation*****
We have made significant improvements to our presentation, especially regarding the positioning and advantage of our model, the relation to multihead attention, comparison with related work, and introduction of the model.  More concretely:
1. In introduction, we highlighted the uniqueness and advantage of AFT, which is a new family of model that has linear space and time complexity w.r.t. both sequence length and feature dimensions. Table 1 is added for a direct comparison with relevant works.
2. In methodology, we changed the way AFT is presented. We have made it clear that we introduce AFT as a new family of model,  but with connections with MHA in one extreme case. We have also made direct contrast with Linear Attention [1] to highlight the relationship and differences.
3. Related work section is reorganized, with more references while emphasizing on the distinction of AFT with groups of existing works.

*****Improved and enriched experiments*****
We have also made substantial efforts trying to make the experiments more convincing. A few highlights are:
1. We added Figure 2 & 3, highlighting the runtime and memory efficiency of AFT during training time, as well as its efficiency during decoding time. These results serve as the empirical counterpart of Table 1.
2. Figure 4 is updated by adding the new sigmoid-softmax nonlinearity, which outperforms our original relu-softmax design.
3. Table 2 includes more model variants, with greatly improved performance. Controlled benchmarking experiments is added to compare with the Image Transformer baseline, where we show great advantage wrt performance, speed and memory.  Discussions are updated to reflect this change.
4. We have removed the wikitext103 experiments as we were not able to complete the experiments due to compute resource constraints. We have instead added Enwik8 which is a popular character level language model benchmark. We have performed extensive experiments, comparing to the standard Transformer, as well as Reformer, Linear Transformer and Sythesizer.  We have also provided ablation studies for our model variants. Check Table 3&4 and related discussions for details.
5. The MT benchmarking is redone with our optimized implementation, where we now show similar speed but better memory efficiency compared to the standard Transformer.
6. Image super resolution experiments are moved to the appendix.

References:
[1] Transformers are RNNs: Fast Autoregressive Transformers with Linear Attention, Katharopoulos et al.

---

### Decision · Program_Chairs · 2021-01-07
**Final Decision**

**Decision:**

Reject

**Comment:**

The new non linearity proposed in this paper present interesting observations and improvements on image and text datasets.
However, reviewers point out that there should’ve been more comparisons to other efficient transformers and on more datasets.
The speed improvements are also not clear.
I’d encourage the authors to revise and submit in the future.